# Combustion Process of Canola Oil and n-Hexane Mixtures in Dynamic Diesel Engine Operating Conditions

**Rafał Longwic [1],\*** , **Przemysław Sander [1]** , **Anna Zdziennicka [2]** , **Katarzyna Szymczyk [2]** and **Bronisław Jańczuk [2]**

[1] Department of Vehicles, Faculty of Mechanical Engineering, Lublin University of Technology, 20-618 Lublin, Poland; p.sander@pollub.pl

[2] Department of Interfacial Phenomena, Institute of Chemical Sciences, Faculty of Chemistry, Maria Curie-Sklodowska University in Lublin, Maria Curie-Sklodowska Sq. 3, 20-031 Lublin, Poland; aniaz@hektor.umcs.lublin.pl (A.Z.); katarzyna.szymczyk@poczta.umcs.lublin.pl (K.S.); bronislaw.janczuk@poczta.umcs.lublin.pl (B.J.)

\* Correspondence: r.longwic@pollub.pl; Tel.: +48-606-785-513

**Abstract:** The article discusses the problem of using canola oil and n-hexane mixtures in diesel engines with storage tank fuel injection systems (common rail). The tests results of the combustion process in the dynamic operating conditions of an engine powered by these mixtures are presented. On the basis of the conducted considerations, it was found that the addition of n-hexane to canola oil does not change its energy properties and significantly improves physicochemical properties such as the surface tension and viscosity. It contributes to the improvement of the flammable mixture preparation process and influences the course of the combustion process.

**Keywords:** combustion; alternative fuel; canola oil (Co); diesel engine; common rail; diesel fuel (Df); n-hexane; injection

## 1. Introduction

In general use, the diesel engine will be used for many years to come (road, sea, rail, stationary engines of working machines) [1,2]. However, the sharpening of the standards concerning the emission of toxic compounds and the unstable situation on the market of petroleum-derivative fuels make it necessary to conduct research on the improvement of fuel supply systems and the construction of the internal combustion engine itself. This is done for obvious reasons, as it may have a positive impact on the improvement of the injection and combustion processes, which may have an influence on the reduction of fuel consumption and indirectly on the reduction of exhaust gas emissions. Another direction of research, leading to compliance with rigorous exhaust emission standards, is to improve the properties of fuels used in internal combustion engines. The physicochemical properties of fuels have a direct impact on the course of the injection and combustion processes [3]. Hydrocarbon alternative fuels not derived from crude oil processing, i.e., diesel, gasoline, etc., have an increasing share in the market for diesel fuels: CNG (compressed natural gas); alcohols (methanol, ethanol, and butanol); vegetable oil (canola oil, soya oil, sunflower oil, palm oil, and peanut oil); and methyl esters of vegetable oils such as canola oil and palm oil.

Additives to diesel or vegetable oils, which are intended to improve the combustion process and contribute to reducing the amount of toxic components in the exhaust gases, are also used. The main additives that have been tested are ethanol, methanol, ethers (methyl tertiary butyl, EMTB; ethyl tertiary butyl, EETB; dimethyl, DME; diethyl, DEE), n-butanol, and FAME additives (fatty acid methyl esters).

The power supply of motor vehicles with alcohol fuels (methanol, ethanol, butanol), ethers (ETTB, DME, DEE), or synthetic fuels is not common and is usually limited to vehicles or research engines. The use of alcohol as a fuel is made more difficult by its low auto-ignition capacity, low calorific value, poor blending with diesel, and high hygroscopicity, so a suitable emulsifier would have to be used to obtain stable diesel–alcohol mixtures. Among alcohols, ethanol is the most often used because it has an advantage over methanol in the range of calorific value and octane number, and in a mixture with petrol, it becomes more difficult to stratify (also at lower temperatures). In order to solve this problem, studies are being conducted on the development of microemulsions consisting of insoluble liquids, i.e.,vegetable oil and methanol, ethanol, or ionic/non-ionic amphiphilic compounds [2–17].

Particularly worth mentioning are fuels derived from oil plants (palm oil, coconut oil, canola oil, soybean oil, linseed oil, peanut oil). Plant oils belong to the group of unconventional liquid fuels. Plant fuels and their esters are obtained from plant seeds. In Polish conditions, it is canola oil, which can be used to power a diesel engine. Research works on the use of canola oil as fuel were conducted in one of three ways, i.e., the use of refined canola oil; use of Co mixtures with Df and alcohols; and the use of canola oil methyl ester (FAME, biocomponent for Df).

The concept of using canola oil as a fuel presented in the article assumes supplying vegetable fuel to diesel engines without their structural changes. Due to known problems in the use of canola oil as fuel and the different physicochemical properties of canola oil, a minimum amount of additive (n-hexane) was applied to canola oil, which brings the physicochemical properties of canola oil closer to the physicochemical properties of diesel fuel [18,19]. The chemical additive used in a small amount in the mixture with canola oil enables the use of canola oil as fuel for diesel engines, taking into account the following functions of the purpose: the physicochemical parameters of canola oil mixtures with additions are similar to the physicochemical parameters of diesel oil in a wide range of ambient temperatures, including the maximum energy parameters of the engine operation and the minimum emission value of the toxic components of the exhaust gas.

The use of an n-hexane additive enables canola oil to be used as fuel in the common use for means of transport. In the first phase of the research, physicochemical tests of canola oil mixtures with n-hexane addition were carried out, and the possibility of using the above solution in a group of engines with conventional injection systems, i.e., engines of agricultural tractors, passenger cars, trucks, and stationary engines equipped with various types of in-line and rotary injection pumps [20,21].

In the presented article, the possibility of using canola oil and n-hexane mixtures in compression ignition engines with storage injection systems was examined, and the course of combustion processes in dynamic engine operating conditions was determined.

## 2. Materials and Methods

### 2.1. Fuels Tested

Diesel fuel (Df) complying with EN590 [22], commercial canola oil (Co), and non-reactive solvent n-hexane (Hex), whose main physicochemical properties are shown in Table 1, were used in the tests. N-hexane ($C_6H_{14}$) is an organic chemical compound from the alkane group. N-hexane isomers are very unreactive and are often used as solvents in organic reactions as they are highly non-polar. On the basis of canola oil (Co), two mixtures with n-hexane were prepared in the following proportions (*v/v*): 10% (10%Hex90%Co) and 15% (15%Hex85%Co). For the fuels tested, their basic physicochemical properties are specified in Table 2.

The equilibrium surface tension ($\gamma_{LV}$) of canola oil+n-hexane mixtures (5-20%Hex) was measured by the Krüss K9 tensiometer according to the platinum ring detachment method (duNouy's method) at 293K. Before the surface tension measurements, the tensiometer was calibrated using water ($\gamma_{LV} = 72.8$ mN/m at 293 K) and methanol ($\gamma_{LV} = 22.5$ mN/m at 293 K). The ring was cleaned with distilled water and heated to a red color with a Bunsen burner before each measurement. In all cases, more than 10 successive measurements were performed. The standard deviation was ±0.1 mN/m.

The density of the aqueous solution of studied canola oil+n-hexane mixtures was measured with a U-tube densitometer, DMA 5000 Anton Paar, with the precision of the density measurements equal to $\pm 0.000005$g cm$^{-3}$. The uncertainty was calculated to be equal to 0.01%. The viscosity measurements of the canola oil+n-hexane mixtures were performed with the Anton Paar viscometer, AMVn, with the precision of 0.0001mPas and an uncertainty of 0.3%. All density and viscosity measurements were made at 293 K.

**Table 1.** Physicochemical properties of n-hexane [23].

| Parameter | Unit | Value |
|---|---|---|
| Kinematic viscosity index in 20 °C | Mm$^2$/s | 0.50 |
| Vapour pressure in 20 °C | Mbar | 160 |
| Dynamic viscosity index in 20 °C | mPa·s | 0.326 |
| Density in 20 °C | g/mL | 0.66 |
| Solubility in water in 20 °C | g/dm$^3$ | 0.00095 |
| Ignition temperature | °C | −22 |
| Boiling point temperature | °C | 68 |
| Self-ignition temperature | °C | 240 |
| Melting temperature | °C | −94 |
| Explosiveness limits | % | low: 1.0 obj.; high: 8.1obj. |

**Table 2.** Selected physicochemical properties of the fuels studied [20,22].

| Parameter | Unit | Value | | |
|---|---|---|---|---|
| | | Df | 10%Hex | 15%Hex |
| Density in 15 °C | (kg/m$^3$) | 835 | 895 | 887 |
| Kinematic viscosity in 40 °C | (mm$^2$/s) | 2.7 | 19.6 | 15.2 |
| Cold filter block age temperature | (°C) | −12 | −3 | −7 |
| Ignition temperature | (°C) | 72 | <40 | <40 |
| Surface tension | (mN/m) | 29.2 | 28.4 | 27.0 |

### 2.2. Research Methodology and Research Station

The test object was a diesel engine with a common rail storage injection system installed in a Fiat Qubo vehicle, which met the Euro 5 emission standards. The vehicle was equipped with a five-stage gearbox. The test vehicle was equipped with an additional (external) independent fuel tank with an additional fuel pump, allowing for the quick replacement of the tested fuels. The modification of the fuel system of the vehicle concerned a low pressure system (about 3 bar). After switching to the auxiliary fuel system (low pressure), the main fuel system was automatically disconnected. The high-pressure fuel system was not modified, so the fuel pressure in the high-pressure system was the same for all the tests performed. The technical data of the engine and the view of the test vehicle are shown in Table 3 and Figure 1.

**Table 3.** Technical data 1.3 multijet test vehicle Fiat Qubo [24].

| The Number of Cylinders | 4 |
|---|---|
| Cylinder diameter (mm) | 69.6 |
| Piston stroke (mm) | 82 |
| Total capacity (cm$^3$) | 1248 |
| Maximum power (kW CEE) | 55 |
| Maximum power (HP CEE) | 75 |
| Operating at maximum power (rpm) | 4000 |
| Maximal moment (Nm CEE) | 190 |
| Maximal moment (kgm CEE) | 19.4 |
| Speed at maximum torque (rpm/1 min) | 1500 |
| Idle rotation speed (rpm) | 850 ± 20 |
| Compression degree | 16.8:1 |

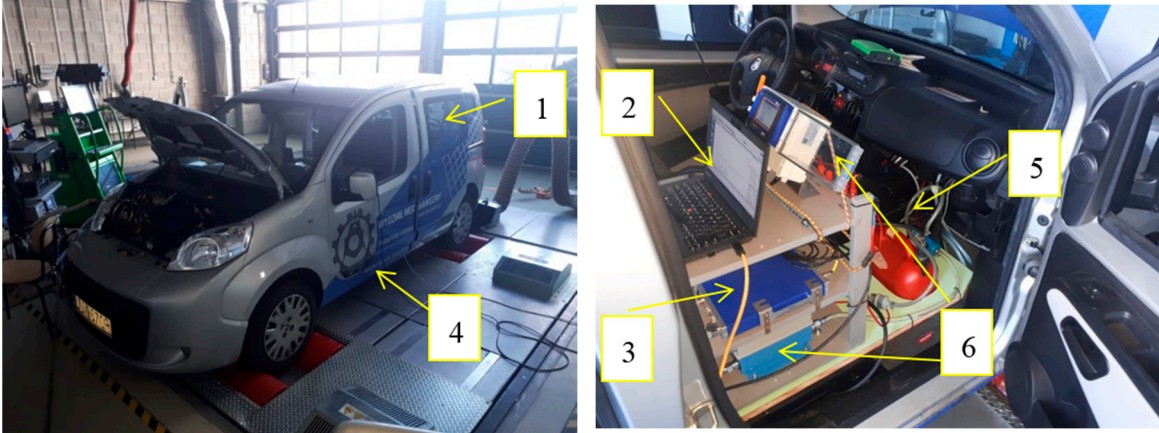

**Figure 1.** Research station simulating vehicle motion under traction conditions: Fiat Qubo test car with 1.3 Multijet engine. 2. Computer with installed AVL Indicom V2.7 software. 3. Indimicro 602 engine indicator system. 4. DF4FS-HLS chassis dynamometer. 5. Additional fuel system tank.6. Rotameter with Multicon CMC-99 module by Simex.

The Indimicro602 recording system of the AVL company with a built-in signal amplifier, cooperating with four analog input channels and two digital inputs, allowing for the recording of quick-change parameters in real time, was used to indicate the engine of the test vehicle. The connection diagram of the system is shown in Figure 2. The signals recorded by the AVL Indimicro system include the following. First, there is the pressure course inside the cylinder, which was recorded by the piezoelectric sensor AVL GH13P installed in the glow plug socket of the first cylinder by means of an adapter (Figure 2, position 3) and whose signal was processed in the amplifier—the AVL measuring module (Figure 2, position 6). Then, the engine crankshaft position signal informing about the crankshaft position was obtained from the induction sensor cooperating with the toothed flywheel by means of the AVL universal pulse conditioner 389Z01 analog–digital converter (Figure 2 position 5).In addition, the injection parameters were analyzed on the basis of the analog control signal of the electromagnetic injector after conversion into a digital signal (Figure 2 item 7).

An indication of a compression ignition engine in conditions simulating vehicle motion in traction conditions was performed by simulating driving on a DF4FS-HLS chassis dynamometer. The diagram of the test stand is shown in Figure 1. The DF4FS-HLS chassis dynamometer was part of the system, which included a stationary (first) roller set with an electro-vacuum brake and hydraulic pump; a mobile (second) roller set with an electro-vacuum brake, hydraulic pump, and gear motor for drive; a control panel (dashboard);control of the hydraulic system; an axial fan for cooling the vehicle; and a PC with dynamometer software.

The combustion engine of the traction unit is exploited mainly under dynamic conditions [25]. Vehicle driving tests are currently used to evaluate the energy and ecological parameters of engines. The most commonly used driving test is the Worldwide Harmonized Light Vehicles Test Procedure (WLTP). The characteristic points of the WLTP driving test corresponding to the dynamic conditions of engine operation have been selected.

Therefore, the tests consisted of a series of measurements for each of the fuels, which allowed the recording of fast-changing parameters during the acceleration of the engine with diesel. First of all, the torque (Mo) and power (Ne) values of the engine that were supplied with the tested fuels were determined. In the next stage of the research, a series of accelerations was performed, during which the following test methodology was applied (the measurement conditions result from the selection of characteristic points of the WLTP test running in dynamic conditions): gearbox ratio—fourth gear, initial vehicle speed 60 km/h, final speed—80 km/h, acceleration lever pitch during testing—40% constant for all fuels. The vehicle was loaded with rolling resistance forces.

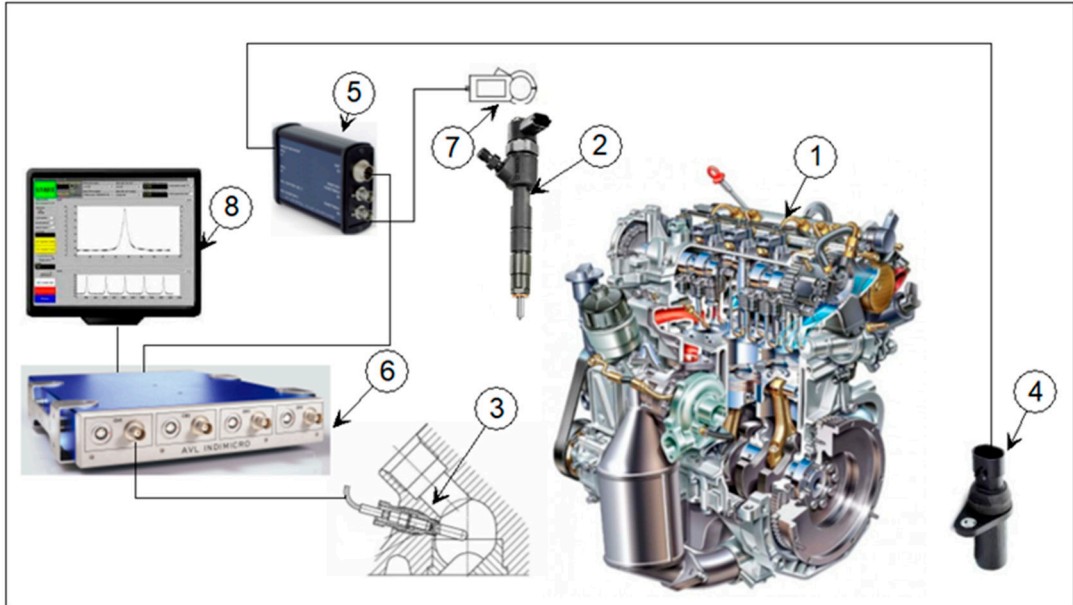

**Figure 2.** AVL connection diagram: 1. Diesel engine of the test vehicle, 2. Electromagnetic injector and cylinder, 3. Piezoelectric sensor AVLGH13P integrated with an adapter for glow plug and cylinder socket, 4. Crankshaft inductive sensor, 5. AVL Universal Pulse Conditioner 389Z01 transmitter, 6. AVL Indimicro 602 measuring module, 7. Measuring rod, 8. Mobile computer with AVL Indicom software.

Selected parameters of the combustion process, including the mean indexed pressure ($p_i$), maximum combustion pressure ($P_{cmax}$), maximum rate of increase of combustion pressure ($dp/d\alpha_{max}$), and the amount of heat produced, were calculated with the use of commonly known laws of thermodynamics, which were implemented in the AVL Indicom software. The calculations were performed at a frequency of $1^0$CA.

In the presented paper, the authors of each fuel performed 10 dynamic acceleration tests of the vehicle, during which the engine operating parameters were recorded. Table 4 presents the values of engine parameters averaged over 10 acceleration tests (at selected engine speed points). Figures 3–8 show representative graphs from the conducted tests.

**Table 4.** Average results of 10 tests of the engine for operating parameters at selected vehicle speeds under dynamic conditions, simulation of extra-urban traffic "High" according to the Worldwide Harmonized Light Vehicles Test Procedure (WLTP) driving test standard.

| Fuel | "I" 60 km/h | | | | "II" 70 km/h | | | |
|---|---|---|---|---|---|---|---|---|
| | N [rpm] | $p_i$ [Mpa] | $P_{cmax}$ [Mpa] | $dp/d\alpha_{max}$ MPa/°CA | n [rpm] | $p_i$ [Mpa] | $P_{cmax}$ [Mpa] | $dp/d\alpha_{max}$ MPa/°CA |
| Df | 1928.6 | 0.95 | 8.71 | 0.458 | 2242.2 | 1.04 | 8.52 | 0.406 |
| 10%Hex90%Co | 1928.6 | 0.953 | 9.05 | 0.314 | 2242.2 | 0.935 | 7.9 | 0.32 |
| 15%Hex85%Co | 1928.6 | 0.915 | 8.06 | 0.488 | 2242.2 | 0.943 | 7.78 | 0.336 |
| | "III" 74 km/h | | | | "IV" 80 km/h | | | |
| | N [rpm] | $p_i$ [Mpa] | $P_{cmax}$ [Mpa] | $dp/d\alpha_{max}$ MPa/°CA | n [rpm] | $p_i$ [Mpa] | $P_{cmax}$ [Mpa] | $dp/d\alpha_{max}$ MPa/°CA |
| Df | 2389.5 | 1.07 | 8.99 | 0.365 | 2509.4 | 1.06 | 9.61 | 0.399 |
| 10%Hex90%Co | 2389.5 | 0.936 | 8.49 | 0.359 | 2509.4 | 0.929 | 9.12 | 0.468 |
| 15%Hex85%Co | 2389.5 | 0.924 | 8.14 | 0.389 | 2509.4 | 0.931 | 8.8 | 0.516 |

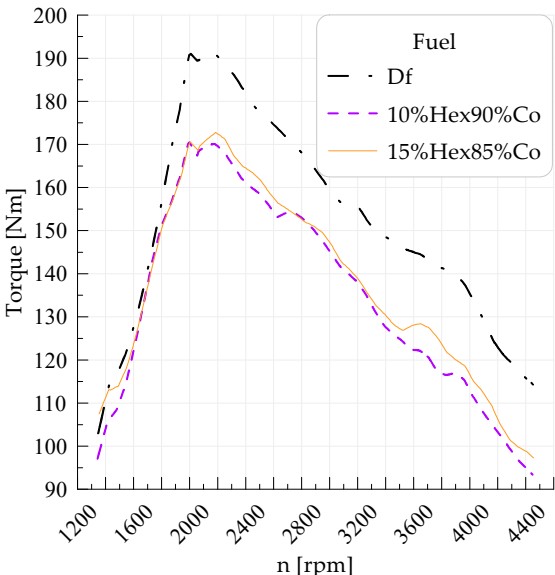

**Figure 3.** Torque values depending on the rotational.

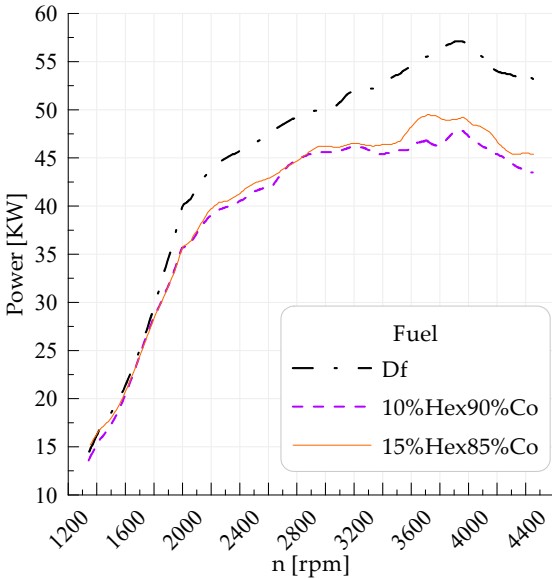

**Figure 4.** Power values depending on the rotationalspeed, engine with diesel supplied with the testedspeed supplied with the tested fuels, i.e., Df, fuels, i.e., Df, 10%Hex90%Co, 15%Hex85%Co, 10%Hex90%Co, and 15%Hex85%Co.

## 3. Results

Maximum torque (Nm) and maximum power (KW) were determined during engine tests. For diesel fuel (Df), the maximum torque and power values were 191.3 Nm at 2123 rpm and 57 kW at 3916 rpm, respectively. For vegetable fuels, lower torque values were obtained of about 11% (10%Hex90%Co) and 10% (15%Hex85%Co) and power of about 17% (10%Hex90%Co) and 14% (15%Hex85%Co). The accelerations obtained were the lowest for the canola oil with a 10% addition of n-hexane. The effect of the amount of n-hexane addition in Co on the obtained values of torque and engine power as well as acceleration, which may be related to the influence of physicochemical properties of fuels, was observed. Detailed values obtained during tests of the test vehicle powered by the tested fuels are presented in Table 5.

During the tests conducted on the diesel engine fueled with the tested fuels, the average acceleration of the test vehicle was obtained, the course of which is shown in Figure 5:Df—0.55 m/s$^2$, 10%Hex90%Co—0.39 m/s$^2$, and 15%Hex85%Co—0.50 m/s$^2$.

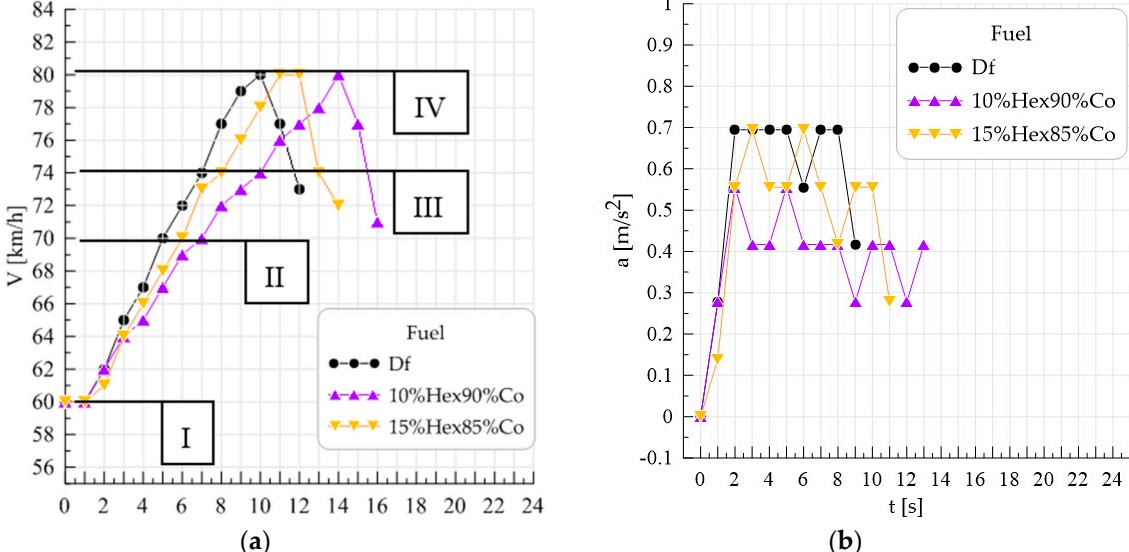

**Figure 5.** The speed of the test vehicle (**a**) and acceleration of the vehicle (**b**) loaded with resistance to movement of the vehicle, powered by the tested fuels, i.e., Df, 10%Hex90%Co, and 15%Hex85%Co. Initial conditions: gearbox ratio—IV gear, initial vehicle speed—60 km/h, final speed—80 km/h, acceleration lever pitch—constant for all fuels—40%. Sampling frequency in 1s.

**Table 5.** The maximum torque and maximum engine power obtained when performing a test on a chassis dynamometer. Df: diesel fuel.

| Parameter | Df | 10%Hex90%Co | 15%Hex85%Co |
|---|---|---|---|
| Power of the vehicle | 57 KW/3916 rpm | 47.8 KW/3995 rpm | 49.5 KW/3714 rpm |
| Torque | 191.3Nm/2123 rpm | 170.5 Nm/1998 rpm | 172.8 Nm/2186 rpm |
| Power on the wheels | 45.5 KW/3893 rpm | 39.2 KW/2891 rpm | 40.3 KW/3703 rpm |
| Maximumspeed | 143km/h/4520 rpm | 143 km/h/4509 rpm | 143 km/h/4516 rpm |
| Acceleration time | 35.30s | 42.88s | 37.98s |

During the performed accelerations of the vehicle, points were selected (shown in Figure 5a) at which combustion process parameters were determined, i.e., the mean index pressure ($p_i$), maximum combustion pressure ($P_{cmax}$), and maximum rate of increase of combustion pressure ($dp/d\alpha_{max}$). Detailed values are shown in Table 4. It was found that the average indexed pressure at each of the measurement points (I, II, III, and IV) was the highest for diesel oil. The biggest difference was observed in relation to 15%Hex85%Co and reacheda maximum of 13%. It was observed that increasing the content of n-hexane in the mixture with canola oil led in most of the cases to a decrease in the mean index pressure and maximum combustion pressure and to an increase in the maximum rate of increase of combustion pressure. Similar trends in the observed parameters were observed for other engine speeds analyzed.

Figure 6 shows the course of the main parameters of the combustion process of an engine fueled with diesel oil (position a), canola oil with 10% n-hexane (position b), and canola oil with 15% n-hexane (position c). It was found that under dynamic conditions, the self-ignition delay angle ($\alpha$ID) was the lowest for Df in the whole engine speed range. For 15% of the n-hexane content in Co, the angle of auto-ignition delay was higher by approximately 13% in relation to Df. Rapeseed fuels were characterized by comparable self-ignition delay values.

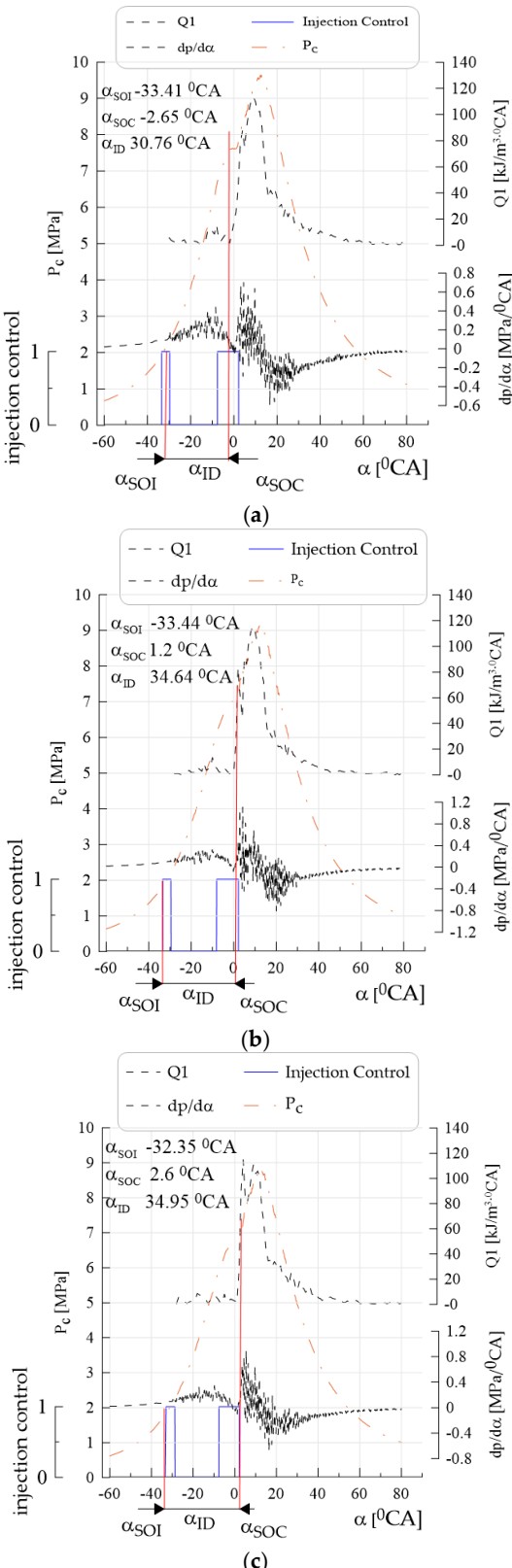

**Figure 6.** The course of the main injection and combustion parameters, i.e., the rate of heat transfer (Q1), pressure inside the combustion chamber (Pc), and the injector control signal—in dynamic engine operating conditions, for a selected engine speed of approximately 2509 rpm (for a speed of 80 km/h). The following fuels were tested: (**a**) Df, (**b**) 10%Hex90%Co, and (**c**) 15%Hex85%Co.Additionally, the following angles were given: αSOI(Start of Injection), αSOC(Start of Combustion), and αID(Ignition Delay). Sampling frequency in 1°CA.

## 4. Discussion

The different physicochemical properties of mixtures of canola oil and n-hexane in comparison with diesel oil, as shown in Table 2, cause different coursesin the injection and combustion processesof these fuels. In diesel engines, one of the most important parameters affecting the combustion process is the self-ignition delay [26]. Differences in the calorific value and the heat of combustion of mixtures cause changes in the operational indicators of engine operation. In the dynamic conditions of engine operation supplied with the tested fuels, the injection time was similar (the differences reached a maximum of 5%). This resulted in the delivery of a volume-comparable amount of fuel to the combustion chamber.

The reason for the lower values of the average indicated pressure (Table 4) that were obtained when supplying canola oil with an addition of n-hexane to Df was the generation of less heat during combustion, which is shown, among others, in Figure 7. During combustion processes of diesel oil on the heat-excitation rate curves, the classical kinetic and diffusion phase could be distinguished, whereas during the combustion of canola oil with the addition of n-hexane, the kinetic phase disappeared with increased n-hexane content, which resulted in higher pressure rise velocities—see Figure 8. This also significantly affected the beginning of combustion, whose angle for diesel oil appeared the earliest (before the kinetic phase of combustion). For canola oil with added n-hexane, the start of combustion appeared later by approximately 4–12°CA in terms of Df(depending on engine speed), while an increase in the amount of n-hexane caused further delay in the start of combustion. A similar tendency was observed forthe angle of self-ignition delay. Figure 6 illustrates the differences in the occurrence of the angle of the beginning of combustion and other parameters of the combustion process when feeding the fuels under study. Another reason for the observed changes could have been a different course of the flammable mixture formation process, due to the different viscosity of the fuels tested. The occurrence of the kinetic phase of combustion resulted in a decrease in the maximum rate of increase of combustion pressure ($dp/d\alpha$) in the engine fueled with Df. For canola oil, with the addition of n-hexane, the maximum rate of pressure rise was higher than for Df, and brewing increased with increase of n-hexane content in the mixture.

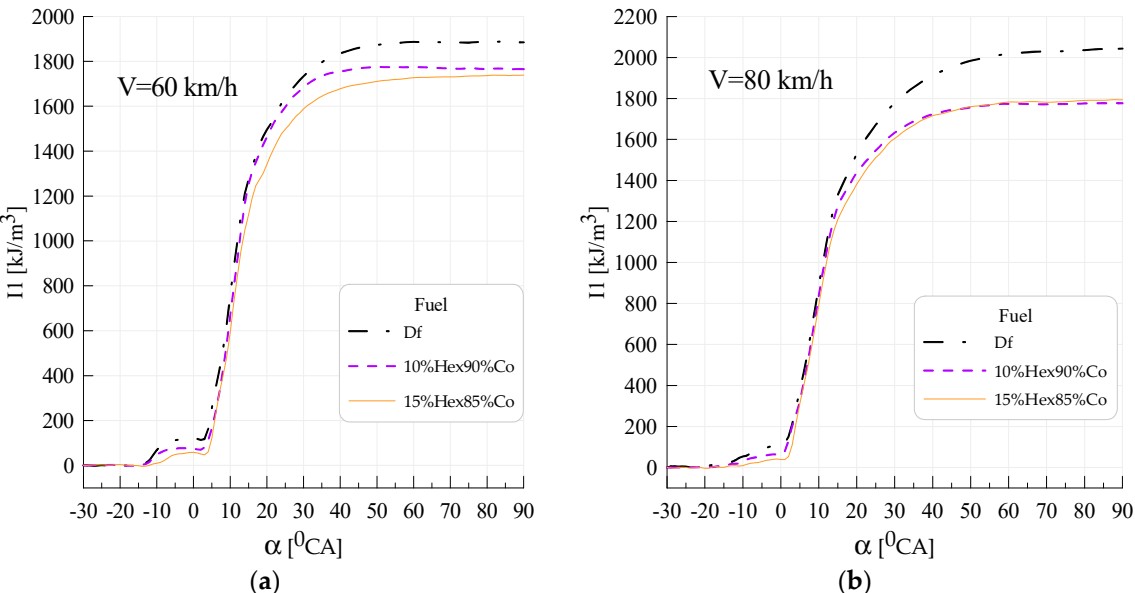

**Figure 7.** Heat discharge (I1) under dynamic engine operating conditions, (**a**) vehicle speed 60 km/h and engine speed approximately 1928 rpm, (**b**) vehicle speed 80 km/h and engine speed approximately 2509 rpm, tested fuels: Df, 10%Hex90%Co, and 15%Hex85%Co. Sampling frequency in 1°CA.

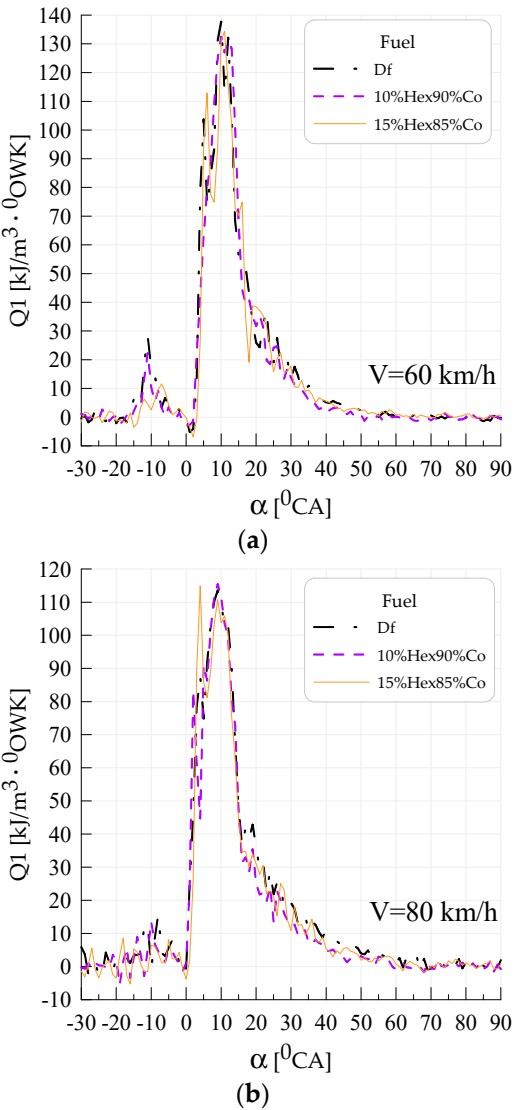

**Figure 8.** Combustion heat transfer rate (Q1) under dynamic engine operating conditions (**a**) vehicle speed 60 km/h and engine speed approximately 1928 rpm, (**b**) vehicle speed 80 km/h and engine speed approximately 2509 rpm, tested fuels: Df, 10%Hex90%Co, and 15%Hex85%Co. Sampling frequency in 1°CA.

The addition of n-hexane to canola oil results in a reduction of surface tension, slight changes in density, and significant changes in viscosity (Figures 9 and 10). The surface tension value of the oil mixture and 5% n-hexane is only 0.8 mN/m higher than the surface tension of the diesel oil. In turn, for the mixture of oil and 10% n-hexane, it is 0.8 mN/m lower. Changes in surface tension as a function of n-hexane concentration in the mixture with canola oil indicate that the mixture does not behave as an ideal mixture (Figure 9) [27], i.e., changes in this tension are not directly proportional to the composition of the mixture. According to the suggestion of Fowkes and van Oss, the surface tension results from Lifshitz-van der Waals interactions, hydrogen bonds, and electrostatic interactions [28,29]. Therefore, it depends on the type of functional groups occurring in the liquid molecules at the phase boundaries. In the case of a mixture of canola oil and n-hexane, the dominant functional groups that may occur at the liquid–air interface are –CH$_3$, =CH$_2$, ≡CH, =CO, and –COOH. The surface tension of the canola oil/n-hexane mixture depends on the density of the mentioned functional groups. Since the main components of canola oil are unsaturated higher fatty acids (>90%), the contribution of the =CO and –COOH groups to the surface tension of the oil is significant. Therefore, the orientation of oleic, linoleic,

and linoleic acid molecules at the phase boundary has a major influence on the surface tension of canola oil. The n-hexane molecules present in the mixture with canola oil due to the strong hydrophobic interactions between the n-hexane molecules and the apolar part of unsaturated fatty acids increase the likelihood of orientation of the formed acid–n-hexane complexes with the hydrophobic part directed toward the gas phase, which in turn lowers the surface tension of the mixture. This may explain the nonlinear relationship between the surface tension of the mixture and its composition (Figure 9).

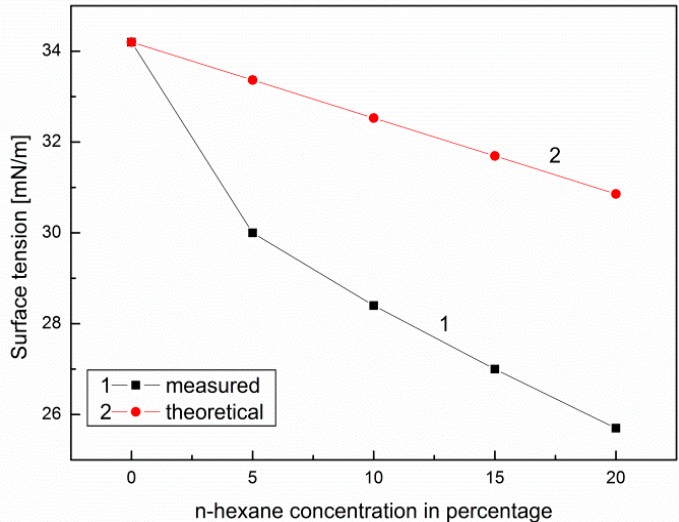

**Figure 9.** A plot of the surface tension of canola oil+n-hexane solution vs. n-hexane concentration. Curve 1 corresponds to the measured values, while curve 2 corresponds to the theoretical values.

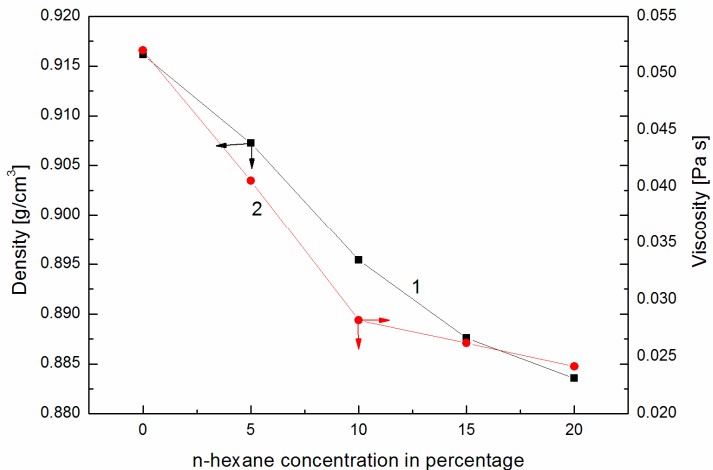

**Figure 10.** A plot of density (curve 1) and viscosity (curve 2) of canola oil+n-hexane solution. The x-axis indicates n-hexane concentration.

The value of surface tension, which is closely related to the interactions between functional groups in the surface layer, can be a decisive factor for the vapor pressure of individual components of a mixture. Since the changes in the surface tension of the tested mixtures as a function of n-hexane concentration indicate non-ideal behaviour of the mixture, it can be expected that the vapor pressure of n-hexane and canola oil components does not comply with the Raoult law [27]. This, in turn, may have a considerable influence on the obtained flash temperature values of the tested blends (<40°C). Moreover, the decrease in surface tension of the canola oil/n-hexane mixture as a function of the n-hexane volume concentration results in a decrease in the volume of the mixture droplets flowing out of the injector tip, which may have a significant impact on the injection process.

The density of the mixture canola oil and 10% n-hexane is slightly lower than the density of diesel oil (Figure 10). Although the addition of n-hexane significantly reduces the viscosity of canola oil, even with its content in the mixture of 20%, the viscosity of the blend is slightly higher than that of diesel oil. On the other hand, the decrease in viscosity affects the change in the angle at the beginning of combustion.

From the point of view of using a mixture of canola oil and n-hexane for diesel engines, it is important that the combustion heat of canola oil under the influence of n-hexane and the oxygen consumption in this process are changed. It is known that canola oil is composed of many chemical compounds; however, the content of oleic, linoleic, and linolenic acid is over 90% [30]. The process of combustion of these acids can be represented by a reaction (1)–(9):

$$C_{17}H_{33}COOH + 25.5O_2 = 18CO_2 + 17H_2O \tag{1}$$

$$C_{17}H_{33}COOH + 16.5O_2 = 18CO + 17H_2O \tag{2}$$

$$C_{17}H_{33}COOH + 7.5O_2 = 18C + 17H_2O \tag{3}$$

$$C_{17}H_{31}COOH + 25O_2 = 18CO_2 + 16H_2O \tag{4}$$

$$C_{17}H_{31}COOH + 16O_2 = 18CO + 16H_2O \tag{5}$$

$$C_{17}H_{31}COOH + 7O_2 = 18C + 16H_2O \tag{6}$$

$$C_{17}H_{29}COOH + 24.5O_2 = 18CO_2 + 15H_2O \tag{7}$$

$$C_{17}H_{29}COOH + 15.5O_2 = 18CO + 15H_2O \tag{8}$$

$$C_{17}H_{29}COOH + 6.5O_2 = 18C + 15H_2O \tag{9}$$

The following chemical reactions (10)–(12) were used to compare the amount of oxygen needed to burn n-hexane:

$$C_6H_{14} + 9.5O_2 = 6CO_2 + 7H_2O \tag{10}$$

$$C_6H_{14} + 6.5O_2 = 6CO + 7H_2O \tag{11}$$

$$C_6H_{14} + 3.5O_2 = 6C + 7H_2O \tag{12}$$

Based on the combustion reaction of 1 mole of n-hexane and the three acids mentioned above, it can be concluded that the amount of oxygen needed to burn n-hexane is much lower than in the case of acids, but such a comparison does not reflect the amount of oxygen needed to burn equal volumes of these substances. Therefore, the number of oxygen moles needed to burn each of these compounds has been calculated by taking into account the density values. If the product of combustion of all compounds is carbon dioxide then the number of oxygen moles needed to burn 1 $dm^3$ of the compound is 80.34, 80.25, 80.43, 80.43, and 72.75, respectively, for oleic acid, linoleic acid, linoleic acid, linolenic acid, and n-hexane. This comparison shows that the number of oxygen moles necessary to burn the same amount of n-hexane is significantly lower than for the three acids mentioned above. These differences should be reflected in the values of heat of combustion of n-hexane and tested acids, and thus canola oil. The heat value of canola oil combustion per 1 $dm^3$ is 35284.32 kJ. It turned out that this value did not differ much from the value for oleic acid (35147.60 kJ). However, the combustion heat of n-hexane, as expected from the amount of oxygen needed for its combustion, is lower and amounts to 31911 kJ. Taking these values into consideration, the heat of combustion of the canola oil/n-hexane mixture as a function of its composition was calculated. The calculations show that in the range of concentrations (from 0 to 20% of n-hexane in the mixture) the heat of combustion slightly decreases from 35284.32 to 34609.66 $kJ/dm^3$ (Figure 11). Changes in the combustion heat of a canola oil and n-hexane mixture are related to the amount of oxygen needed to burn the mixture. Considering that more than 90% of the oil composition is made up of the acids tested, the average value of oxygen mole

needed to burn 1 dm$^3$ of oil was used to calculate the number of oxygen needed to burn 1 dm$^3$ of oil. The calculated amount of oxygen mole needed to burn 1 dm$^3$ of the mixture, similarly to the heat of combustion, changes slightly (from 80.34 to 78.82). On the basis of the conducted considerations, it can be stated that the addition of n-hexane to canola oil slightly changed the heat of combustion and significantly improved the physicochemical properties such as the surface tension and viscosity.

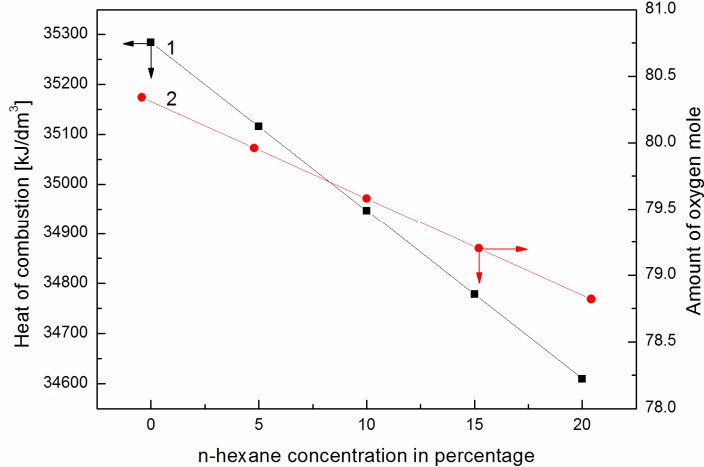

**Figure 11.** A plot of heat of combustion (curve 1) and amount of oxygen mole needed to burn off 1 dm$^3$ of canola oil–n-hexane mixture (curve 2) vs. n-hexane concentration.

The addition of n-hexane to canola oil caused a significant decrease in its viscosity and surface tension. Changes in the surface tension of canola oil with the addition of n-hexane as a function of composition are not linear, and synergy in the reduction of surface tension is observed. It was found that the mixture of canola oil and n-hexane is not ideal. A significant decrease in the viscosity and surface tension of oleic acid, the main component of canola oil, by adding n-hexane, indicates that such a mixture can be used in diesel engines.

## 5. Conclusions

The conducted research and its analysis allow us to formulate a few final conclusions:

- The vapor pressure of n-hexane and canola oil components does not comply with Raoult law, and this in turn may have a significant influence on the obtained values of ignition temperature of the tested blends (<40°C); the reduction of surface tension of the canola oil mixture with n-hexane as a function of n-hexane volume concentration causes a decrease in the volume of mixture drops flowing out of the injector tip, which may have a significant influence on the injection process;
- Based on the calculations carried out, it can be concluded that the addition of n-hexane to canola oil slightly changed the heat of combustion of the same volume of the prepared mixture and significantly improved physicochemical properties such as the surface tension and viscosity;
- Under dynamic conditions of engine operation supplied with the tested fuels, the injection time was similar (the differences reached a maximum of 5%); this resulted in supplying the combustion chamber with a volume comparable amount of fuel;
- Obtaining lower values of the average pressure indexed when supplying canola oil with n-hexane in relation to Df was caused by lower heat generation during combustion; during diesel combustion, the classical kinetic and diffusion phase could be distinguished on the heat transfer rate curves, whereas during the combustion of canola oil with n-hexane, the kinetic phase disappeared with increased n-hexane content, which resulted in higher pressure increase rates; for canola oil with n-hexane addition, the beginning of combustion occurred later by approximately 4–12°CA with respect to Df (depending on engine speed), while an increase in the share of n-hexane caused

further delay in the beginning of combustion—a similar tendency was observed for the angle of self-ignition delay.

**Author Contributions:** R.L., P.S., A.Z., K.S. and B.J. designed the experiments, analyzed the experimental data, made figures, participated in the preparation of the manuscript, and wrote the part of the manuscript. R.L., P.S. and B.J. conceived the concept of the studies, wrote the main part of the manuscript, supervised the studies, participated in the manuscript preparation, signed the experiments, analyzed the experimental data, and made figures. All authors have read and agreed to the published version of the manuscript.

**Funding:** This project/research was financed in the framework of the project Lublin University of Technology-Regional Excellence Initiative, funded by the Polish Ministry of Science and Higher Education (contract No. 030/RID/2018/19).

**Conflicts of Interest:** The authors declare no conflicts of interest.

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
