# Peer review of "Combustion Process of Canola Oil and n-Hexane Mixtures in Dynamic Diesel Engine Operating Conditions"

_applsci, doi:10.3390/app10010080_

Round 1

Reviewer 1 Report

This paper represents the experimental results of the diesel dynamics operation with three different test fuels: 1. diesel fuel, 2 canola oil with 10 % of n-hexane and 3. canola oil with 15% of n-hexane. The authors have presented the experimental results related to combustion cylinder pressure trace, vehicle acceleration performance, and the tested fuels physical-chemical properties. 

All in all the paper is easy to read and the logical is smooth. But some major issues still exist and need the authors' attention.

how many runs that the authors have been tested for each scenario. It seems each test only runs once and if so, the results are not convincible. The authors have used an additional fuel tank in the test. So have you shut down the main fuel supply in the vehicle? What is the pressure for the additional fuel tank as the fuel supply? Is the pressure fixed for all tests? In Figure 4 caption, what does the engine no. 2 mean? For the physical and chemical properties of the mixture fuel, do the authors measured or calculated by the theory model. If measured, please provide details information about the measurement process. The conclusion parts are full of uncertainties, like "which may have a significant influence on the injection process". As a scientific paper, the conclusions can not be like that. Please give us the most significant conclusions with solid evidence.  In your conclusion part, there is a duplicated sentence in line 381. the n-hexane in the keywords is a typo Table 1 title should be with the table  The number indicates each reaction should be in a straight line.

Author Response

Please see annex

Reviewer 2 Report

In this paper, the authors have proposed using canola oil mixed with hexane and apply in dynamic diesel engines, to examine whether the performance can be enhanced, as well as reviewing the physiochemical properties related to such engines. The concept and the presented results in this article is quality. However, it appears that more in-depth analysis and representation of the internal combustion behaviour can be delivered to strengthen the hypothesis of this proposed research. It is, therefore, recommended to be "majorly revised" before the consideration of publishing in MDPI Applied Science. The following are the comments:

It would be preferable if the author can provide more illustration of the spray pattern and combustion flow behaviour in addition to just showing the operating performance. There are two many bullet points appearing in Introduction. For professional writing, it is recommended for the authors to explain their points in a structured paragraph manner. Please revise. The authors should show the data points discreetly in points in Figs. 3-8 and try to avoid using the same line style, since they cannot be separated when printing in black and white. Please revise. In the methodology, what are the equations or algorithms applied to acquire your results (i.e. combustion heat rate, surface tension, density and viscosity). Is there any post-processing techniques applied? Please further elaborate. Please revise the conclusions and avoid using bullet points. The conclusion should be a concise summary of all the key findings with provided future prospectives.

Author Response

Please see annex

Reviewer 3 Report

Based on the submitted article, there are issues that will need to be addressed before being accepted for publication as the English writing will need to be revised and some of the flaws as seen were in the use of; is, are, was, were, the or not adding 'the' and also include spelling of words or their uses inappropriately. Some issues as seen are detailed here; 

Line 34 - is, Line  61 - a fuel, Line 79 - were, Line 84 - combustion, Line 85 - was, Table 1 - Kinemaic, Table 2 - Boiling point temperature :, Ignition temperaure, Table 3 Numer, Line 120 - was, Line 137 - energy, Line 141 - the test therefore (add commas appropriately), Line 251 - addition, Line 264 - indicate, Line 292 - does, Line 304 - are, Line 326 - has, Line 327 - is, Line 339 - are, Line 353 - are, Line 352 - addition, Line 363 - mixture, Line 382 - addition, Line 384 - in.

Also, the names of authors at the top of the paper should be separated differently using commas. At the reference section, the should be uniformity in the writing. The names should be written in the same format rather than writing in block capital in some and others in small letter.

Author Response

Please see annex

Round 2

Reviewer 1 Report

The authors have addressed my comments and the paper's quality has been improved significantly. I suggest accepting this paper in the presented format. 

Author Response

Reply to the Academic Editor's remark.

Thank you very much for the time invested in improving our manuscript.

"Moderate English changes required"

The revised manuscript was checked by the English teacher and we hope that it satisfies your requirements

Kind regards

Rafał Longwic

Reviewer 2 Report

Overall, the authors have addressed most of the comments and the article now is up to publication standards. However, it is recommended that a minor-spelling check should be performed to perfect the manuscript before publication procedures.

Author Response

Reply to the Academic Editor's remark.

Thank you very much for the time invested in improving our manuscript.

" English language and style are fine/minor spell check required"

The revised manuscript was checked by the English teacher and we hope that it satisfies your requirements

Kind regards

Rafał Longwic